# Automatic and Accurate Extraction of Sea Ice in the Turbid Waters of the Yellow River Estuary Based on Image Spectral and Spatial Information

**Huachang Qiu** [1,2] , **Zhaoning Gong** [1,2,*] , **Kuinan Mou** [1,2], **Jianfang Hu** [1,2], **Yinghai Ke** [1,2] **and Demin Zhou** [1,2]

1    College of Resource Environment and Tourism, Capital Normal University, Beijing 100048, China; 2190902141@cnu.edu.cn (H.Q.); 2190902184@cnu.edu.cn (K.M.); 2190902115@cnu.edu.cn (J.H.); yke@cnu.edu.cn (Y.K.); zhoudemin@cnu.edu.cn (D.Z.)
2    MCA Key Laboratory of Disaster Assessment and Risk Prevention, Capital Normal University, Beijing 100048, China
*    Correspondence: gongzhn@cnu.edu.cn

**Abstract:** Sea ice is an important part of the global cryosphere and an important variable in the global climate system. Sea ice also presents one of the major natural disasters in the world. The automatic and accurate extraction of sea ice extent is of great significance for the study of climate change and disaster prevention. The accuracy of sea ice extraction in the Yellow River Estuary is low due to the large dynamic changes in the suspended particulate matter (SPM). In this study, a set of sea ice automatic extraction method systems combining image spectral information and textural information is developed. First, a sea ice spectral information index that can adapt to sea areas with different turbidity levels is developed to mine the spectral information of different types of sea ice. In addition, the image's textural feature parameters and edge point density map are extracted to mine the spatial information concerning the sea ice. Then, multi-scale segmentation is performed on the image. Finally, the OTSU algorithm is used to determine the threshold to achieve automatic sea ice extraction. The method was successfully applied to Gaofen-1 (GF1), Sentinel-2, and Landsat 8 images, where the extraction accuracy of sea ice was over 93%, which was more than 5% higher than that of SVM and K-Means. At the same time, the method was applied to the Liaodong Bay area, and the extraction accuracy reached 99%. These findings reveal that the method exhibits good reliability and robustness.

**Keywords:** Yellow River Estuary; turbid area; spectral information; textural features; sea ice extension; automatic extraction

## 1. Introduction

Sea ice refers to saltwater ice that is directly frozen from seawater, and also includes continental glaciers (icebergs and Iceland), river ice, and lake ice that enter the ocean. Sea ice greatly inhibits the heat and steam exchange between the ocean and atmosphere and alters the radiation budget and energy balance of the ocean's surface. These changes have an important impact on oceanic hydrological conditions, atmospheric circulation, and ocean climate [1]. In addition, the production and disappearance of sea ice greatly affects human marine activities. For example, sea ice significantly affects the development of marine resources and marine transportation [2]. Moreover, sea ice presents a potential freshwater resource [3]. Therefore, accurate real-time monitoring of sea ice bears an important application value and theoretical significance.

The bottom of the Yellow River Delta contains a small flat slope, shallow water low salinity, as a result of which seawater is easily frozen [4]. The development of conditions conducive to ice formation and persistence in this sea area is unstable; that is, the ice disappears as the temperature rises and reappears as the temperature drops [5]. Sea ice

is formed at a rapid rate and responds more closely to the local climate. Sea ice disasters frequently occur in the waters of the Yellow River Delta, and sea ice often has a major impact on fishing ports, wharves, shallow beach aquaculture, and offshore infrastructure [6]. For example, the operation area of the Shengli Oilfield, China's second-largest oilfield, is mainly concentrated in the Yellow River Delta, the Bohai Sea and its adjacent waters. Sea ice severely affects and threatens offshore oilfield production operations and various engineering facilities in the winter [7]. According to statistics, severe and relatively serious sea ice disasters occur roughly once every five years in China, and in some sea areas, sea ice disasters occur almost every year [8]. In the winter of 2009–2010, the Bohai Sea and the northern part of the Yellow Sea experienced the worst ice conditions in nearly 30 years. The severe ice conditions had a significant impact on the society and economy of the provinces (cities) along the Yellow Sea and the Bohai Sea. According to statistics, sea ice disasters caused economic losses totaling nearly USD 900 million [9]. Therefore, in the context of global warming, it is entirely possible that continuous low temperatures and severe sea ice disasters will occur in some areas. In particular, with the rapid economic development of the Bohai Rim region, the losses suffered by sea ice disasters are increasing year by year [10]. In the new era, the country should promote the understanding of the ocean, rationally develop and utilize marine resources, protect the rights and interests of the ocean, and insist on the harmonious coexistence of man and the ocean. Faced with the new requirements as China enters a new era and accelerates the building of its maritime power, the existing sea ice disaster prevention and mitigation capabilities can no longer fully meet the actual needs of the economic and social development in icy sea areas, and sea ice monitoring capabilities remain relatively weak [11]. It is the trend of future development to improve the three-dimensional marine disaster observation network that combines coastal observation, offshore platform and satellite remote sensing, so as to improve the ability of marine disaster observation [12]. Therefore, the precise extraction of sea ice provides the basis for strengthening the analysis of and research on sea ice conditions, and is necessary for improving early sea ice warning technology, as well as ice prevention and disaster reduction capabilities.

Traditional sea ice monitoring methods, such as shore-based observations and ice-breaker observations, cannot obtain large-scale sea ice information in a timely and accurate manner. Remote sensing monitoring technology has a high timeliness, can obtain repeated observations on a large scale, and is relatively inexpensive, providing long-term data support for dynamic and efficient sea ice monitoring [13–15].

Large-scale monitoring of sea ice in high latitudes can be carried out using remote sensing technologies such as microwave and optical remote sensing. Passive microwave and synthetic aperture radar (SAR) imagery enables all-weather observations and the ability to penetrate through clouds [16,17]. The automatic segmentation algorithm based on statistical distance realizes the classification of C-band fully polarized sea ice data [18]. A sea ice classification method is proposed for X-band, C-band and L-band fully polarized synthetic aperture radar images. By extracting the polarized features of sea ice classification, the feature vector provides input into the neural network classifier to realize the extraction of sea ice [19]. However, it is challenging to obtain due to the high cost and the long revisit period for most of them [20]. Optical remote sensing data, although limited by weather conditions, usually delivers better spatial resolution, lower cost, and shorter revisit times. For example, satellites such as Moderate Resolution Imaging Spectroradiometer (MODIS) [21], Advanced Very High-Resolution Radiometer (AVHRR), Geostationary Ocean Color Imager (GOCI) [22], and Feng Yun 3 (FY-3) [23] has a high time resolution and can be used for continuous monitoring of sea ice in a large-scale area and a long time sequence. The disadvantage is that the spatial resolution is low, and it is difficult to perform refined regional sea ice monitoring. High spatial resolution remote sensing data represented by GF1 [24], Landsat [25], and Sentinel-2 [26] can be used to achieve refined sea ice monitoring through data mining, and the effective combination of multi-source medium and high-

resolution satellite data compensates for the low time resolution of the data and further improves its sea ice monitoring capabilities.

Three main types of feature parameters are used for sea ice extraction: spectral features, spatial features, and temperature features. The spectrum processing methods commonly used for sea ice extraction include the band ratio, band difference, and various normalized indexes, which highlight the sea ice information [27]. In addition, the sample point selection and machine learning methods have also been used to extract spectral information about sea ice. The second concerns the spatial features. The spatial feature commonly used for sea ice extraction is the textural feature based on the gray-level co-occurrence matrix. Generally, the surface of sea ice is rough and has conspicuous irregular and unstable textural characteristics; while the surface of sea water is smooth and has constant textural characteristics [28]. The third refers to the temperature feature. This parameter is used to retrieve the surface temperature through the thermal infrared band of the remote sensing image. The temperature feature is simple and easy to use, but has fewer data sources and a low spatial resolution; the accuracy of the temperature retrieval algorithm is also not very high, limiting its application [29].

Current sea ice extraction methods can be divided into threshold segmentation methods, machine learning methods, and digital image processing methods. The threshold segmentation method mainly involves setting the threshold value of the sea ice's spectral, textural, temperature, and other parameters and determining the threshold values using an artificial or bimodal histogram, scatter plots, and other methods, such as using the red band and the ratio threshold of the near-infrared band to achieve rapid extraction of sea ice [27]. Su et al. used the red and near-infrared bands of Sentinel-3 images to establish a sea ice information index that highlights the spectral information of sea ice, and employed the Jenks method to determine the segmentation threshold of ice water [30]. Hayashi et al. used reflectance scatter plots of MODIS bands 1 and 2 to derive a formula suitable for extracting the area of thin ice [31]. Ice conditions in the Gulf of Riga in the Baltic Sea were counted by a bimodal histogram method, the statistical results were limited by the spatial resolution of MODIS [32]. The threshold segmentation method is simple and fast, but it is difficult to determine the threshold value using this method, and the threshold value of different images will be slightly different. Machine learning methods such as support vector machines and Classification and Regression Trees (CART) decision trees select certain sample points and then classify the sea ice and seawater areas [33]. The machine learning method is simple and easy to use, but the classification process is unclear, and the sample points need to be manually selected, making it difficult to achieve automation. In addition, the final accuracy depends entirely on the selected sample points. The digital image processing method is to highlight the sea ice information by processing and transforming the image. For example, Li et al. proposed a linear spectral decomposition method based on MODIS images with multiple constrained end members [34]. The pixels are decomposed to extract the range of the sea ice. Liu et al. used a wavelet transform to extract the textural information from a SAR image, converted a China–Brazil Earth Resources Satellite (CBERS-02B) optical image from Red-Green-Blue space to Hue-Saturation-Intensity space, and finally employed Principal Component Analysis (PCA) to fuse the HSI image and the texture images [35]. Digital image processing methods can highlight sea ice information more intuitively, but the process is more complicated and difficult to automate.

The seawater turbidity of the Yellow River Delta is high, and the changes are quite drastic, which intensifies the distinction between sea ice and seawater. Aiming at the problems of the low accuracy and efficiency of the current sea ice extraction methods used in the Yellow River Delta and based on multi-source remote sensing images, in this study, a set of sea ice automatic extraction method systems suitable for the Yellow River Delta was developed and the spatial and textural information about the sea ice was fully excavated. The automatic extraction method will be further extended to other areas such as Liaodong Bay in order to provide important technical support for the rational development of sea ice resources and to improve the early warning capabilities of sea ice disaster prevention and

mitigation systems. This method explores the spectral information and texture information of different ice types. The spectral information, texture information and edge information of sea ice are used to extract sea ice, and the completeness and accuracy of sea ice extraction results are improved through multi-scale segmentation of images. This method can provide a reliable method for extracting sea ice extent from high-resolution optical remote sensing data such as GF1.

## 2. Materials and Methods

### 2.1. Study Area and Data

The Yellow River Delta is located in the southeastern part of Bohai Bay and the northwestern part of Laizhou Bay (Figure 1). The sea area is located in a typical monsoon climate zone. The winter is controlled by the Eurasian continental high pressure system. The north and northwest winds are dominant, and the weather is dry and cold. When strong cold air invades, it is often accompanied by processes such as strong winds, snowfall, and sharp temperature drops. The coastline is more than 100 km long, and the sea water near the coast mostly contains suspended sediment from the mouth of the Yellow River. Therefore, the sea area has a wide intertidal bandwidth, a small and flat bottom slope, and shallow water depths. In winter, this area is greatly affected by the meteorological conditions and the continent. In addition, affected by the runoff from the Yellow River and other rivers entering the sea, the salt content of the seawater in this sea area is relatively low. The aforementioned special geographical environment and climatic conditions provide sufficient and necessary conditions for the freezing of seawater in this sea area. Sea ice often has a significant impact on fishing ports, wharves, shallow beach aquaculture, and marine infrastructure in this area. With the gradual expansion of the economic scale of the marine aquaculture industry in this area, the impact of sea ice on production activities has become more significant.

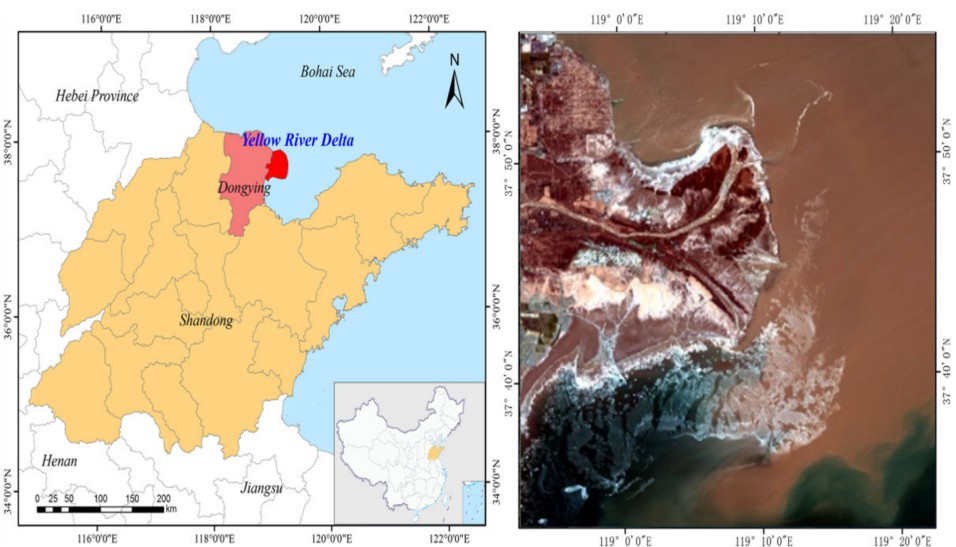

**Figure 1.** The study area.

According to the development stage of sea ice, the sea ice in the Yellow River Delta can be divided into new ice (NI), ice rind, nilas (NL), grey ice (GR), grey-white ice (GW), and white ice. New ice is formed by direct freezing of seawater or snow falling when the temperature is low, and the sea surface is not melted. It is mostly needle-like, flake-like, grease-like, or sponge-like. Ice rind is formed by the freezing of new ice or direct freezing of the calm sea surface. The surface of the ice crust is smooth, moist, and shiny. Its thickness is about 5 cm. It can fluctuate with the wind and is easily broken by wind and waves. Nilas ice is a thin, elastic ice crust with a thickness of less than 10 cm. It easily bends and breaks under external forces and can produce a finger-like overlapping phenomenon. Grey ice is

an ice cap layer with a thickness of 10–15 cm. It is developed from nilas. The surface is flat and moist. It is mostly grey. It is less elastic than nilas ice. It is easily broken by swells and overlaps when squeezed. Grey-white ice is an ice layer with a thickness of 15–30 cm, and is developed from grey ice. It has a rough surface, is greyish-white, and mostly forms when ice ridges are squeezed. White ice describes an ice layer with a thickness of greater than 30 cm. It develops from grey-white ice, has a rough surface, and is mostly white [36].

In this study, Sentinel-2 and Landsat8 images from 2017–2019 were selected to verify the applicability of the method. The time phase and image quality information obtained from the data is presented in Table 1.

**Table 1.** Image information table.

| Area | Date | Image | Band Number | Resolution | Cloud Cover |
|---|---|---|---|---|---|
| Yellow River Delta | 21 January 2017 | GF1 | 4 | 16 m | 1% |
| Yellow River Delta | 12 January 2018 | GF1 | 4 | 16 m | 1% |
| Yellow River Delta | 12 January 2018 | Sentinel-2 | 10 | 10 m | 0% |
| Yellow River Delta | 23 January 2019 | Landsat8 | 7 | 30 m | 0% |
| Yellow River Delta | 21 January 2017 | Planet | 4 | 3 m | 1% |
| Yellow River Delta | 12 January 2018 | Planet | 4 | 3 m | 2% |
| Yellow River Delta | 23 January 2019 | Planet | 4 | 3 m | 1% |
| Liaodong Bay | 17 February 2019 | Landsat8 | 7 | 30 m | 0% |
| Liaodong Bay | 17 February 2019 | Planet | 4 | 30 m | 0% |

The GF1 data used in this article were obtained from the China Resources Satellite Application Center (http://www.cresda.com/CN/, accessed on 21 January 2017). The PIE-Basic software was used for geometric correction, atmospheric correction, orthorectification, image clipping, and other pre-processing work. The Sentinel-2 data were obtained from the European Space Agency's (ESA) data sharing website (https://scihub.Coppe-rnicus.eu/, accessed on 12 January 2018). The Sentinel-2 data released is a product of the Top-of-Atmosphere (TOA) reflectance that has been geometrically corrected and radiometrically corrected, so it was only necessary to perform atmospheric correction of this dataset. The SNAP software officially provided by the ESA was used to perform the atmospheric correction on the downloaded data. The Landsat data were obtained from https://landlook.usgs.gov/ (accessed on 23 January 2019) and the ENVI software was used to perform the FLAASH atmospheric correction. In order to facilitate the subsequent statistical analysis, calculations, and other operations and to reduce the data storage space, all of the reflectance data were expanded by 10,000 times and rounded.

*2.2. Sea–Land Separation*

In order to avoid interference from land information, the sea and land need to be separated before the sea ice extraction. The Normalized Difference Vegetation Index (NDVI) and the Normalized Difference Water Index (NDWI) are the most commonly used indexes for water and land separation. They both use the normalized ratio of the reflectance between visible and near-infrared light. The difference is that the NDVI uses the green and near-infrared bands; while the NDWI uses the red and near-infrared bands. In the Yellow River Delta, the concentration of suspended sediment in some of the seawater is extremely high, which improves the reflectivity of the seawater in the near-infrared band. Figure 2 shows the spectral curve of the seawater and sea ice in the GF1 image. The reflectivity of the clean seawater is significantly higher in the visible light range than in the near-infrared band, while the reflectivity of the highly turbid seawater is higher in the near-infrared band than in the blue-green band. However, it is still lower than the reflectivity in the red band since the reflectivity in the blue and green bands will gradually become saturated as the concentration of suspended sediment increases, while the reflectivity in the red and near-infrared bands will continue to increase as the concentration of suspended sediment increases [37]. However, the near-infrared reflectivity is always lower than the red-band reflectivity. Sea ice 1 and sea ice 2 are defined as sea ice in clean water and turbid water, respectively, and the reflectivity in the near-infrared band is also lower than that in the

red band. Therefore, the effect of using the NDVI for sea–land separation is better. Pixels with NDVI values of less than 0 are classified as seawater, and pixels with NDVI values of greater than 0 are classified as land. In addition, after separation, it is necessary to filter out discrete areas such as rivers and lakes on the land and ultimately to only retain the ocean area. The results of the sea–land separation are shown in Figure 3.

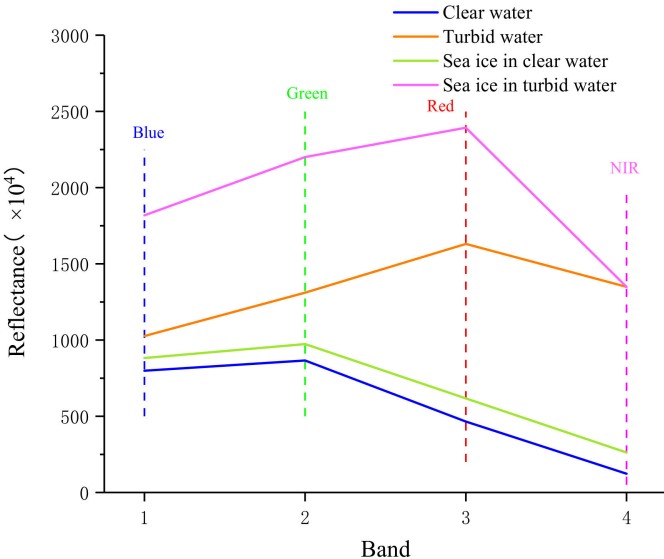

**Figure 2.** The reflectance of the ice and water in the GF1 image.

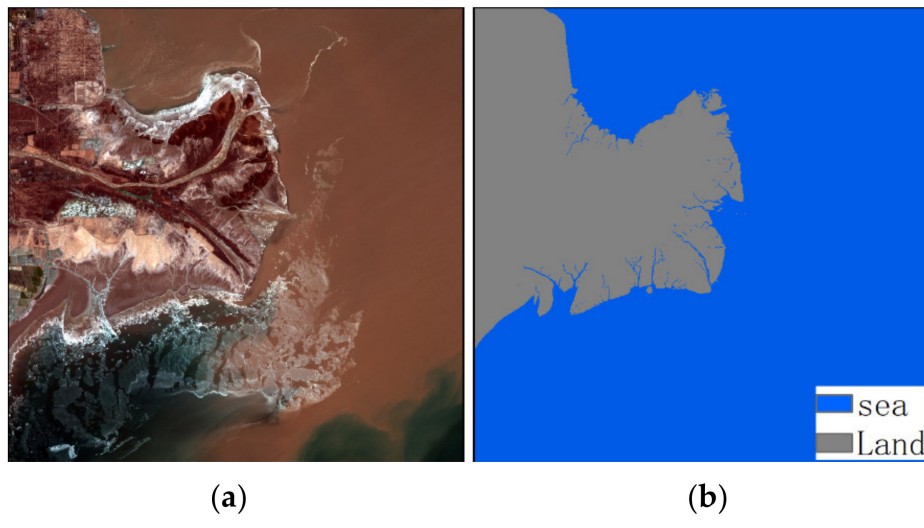

**Figure 3.** (**a**) True color GF1 image acquired on 12 January 2018; and (**b**) the results of the sea–land separation.

### 2.3. Sea Ice Spectral Information Extraction

Spectral information is the most commonly used feature for extraction. In this section, a sea ice spectral information index suitable for different suspended particulate matter concentrations based on the spectral characteristics of sea ice is developed. Generally, the reflectance of seawater in clear seas is higher than that of seawater. However, the suspended particulate matter (SPM) in the Yellow River Estuary increases the reflectance of the seawater. The reflectance of seawater with a high SPM content is even higher than that of sea ice with a lower SPM content (Figure 4). As can be seen from the box plot in Figure 4, as the wavelength increases, the reflectivity of the seawater fluctuates more widely, and the blue band is relatively less affected by the SPM content. This is due to the fact that as

the SPM content increases, the blue band reaches saturation first, followed by the green and red bands. From the perspective of the type of sea ice, in addition to grey ice and grey-white ice, a considerable part of the new ice and ice rind has the same reflectance as seawater. This is because grey ice and grey-white ice are thicker and have a much higher reflectivity than seawater, whereas new ice and ice rind are thin ice with a thickness of less than 10 cm and have a lower reflectivity. Therefore, a single waveband cannot distinguish all types of sea ice from seawater. The difficulty of sea ice extraction in the Yellow River Estuary is mainly the extraction of the new ice and ice rind from seawater with different SPM contents.

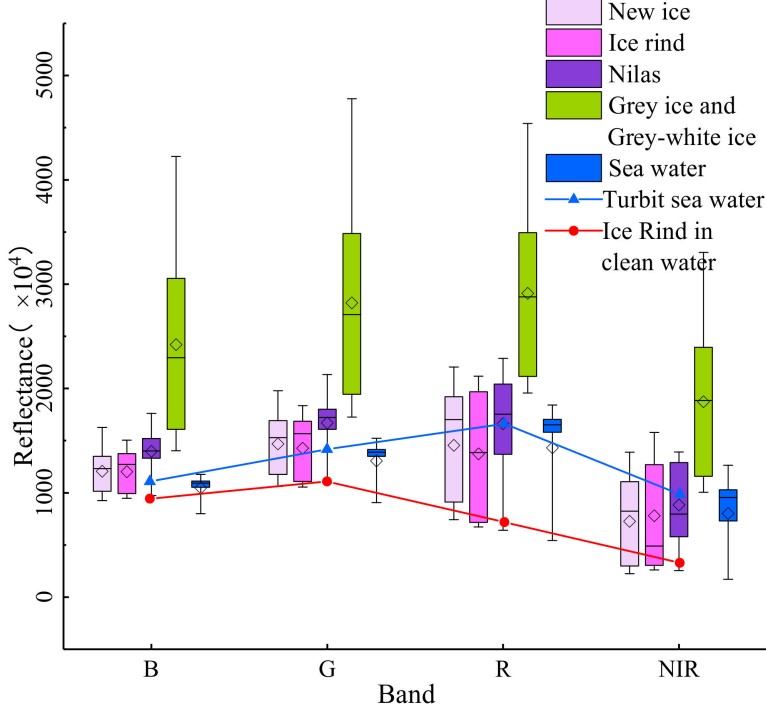

**Figure 4.** Sea ice and water reflectance box plot.

The commonly used spectral index such as the NDVI and NDSI, cannot eliminate the influence of SPM on sea ice extraction. Therefore, in this study, the spectral information concerning the sea ice was extracted by searching for an optimal band combination method. Since the visible light and near-infrared bands are the most important bands for extracting sea ice, these four bands were selected as the best band combination from the data source. The scatter plot can intuitively reflect the separation of the different samples. As shown in Figures 3 and 4, each point in the scatter plot formed by any two bands represents the position of the sea ice or seawater in this two-dimensional space. If the points of sea ice and seawater are scattered together (Figure 5a), then the separation of the sea ice and seawater in the two-dimensional space formed by these two wavebands is poor; and if the sea ice and seawater each gather in one location, it indicates that the sea ice and seawater can be separated well using this waveband combination (Figure 5b). In this way, a straight line separating the sea ice and seawater can be drawn in this two-dimensional space.

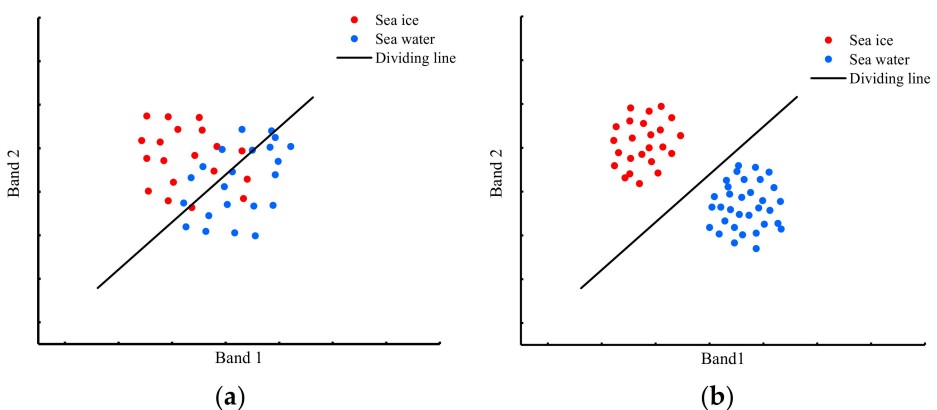

**(a)**　　　　　　　　　　　　　　　　　**(b)**

**Figure 5.** (**a**) Band 1 and Band 2 exhibit good separation in this scatter plot; (**b**) Band 1 and Band 2 display poor separation in this scatter plot.

In order to test all of the band combinations as much as possible, in this study, combinations of the four red, green, blue, and near-infrared bands were tested first. The results of these combinations are presented in Table 2. There are 28 results in total. Two of these 28 bands were chosen to construct a two-dimensional scatter plot, with a total of 378 combinations. For example, if the NDVI can better distinguish between sea ice and seawater, then the scatter plots constructed with the NIR − R and NIR + R bands will exhibit better separation. Finally, the reflectivity of grey ice and grey-white ice is much higher than that of primary ice and ice skins, and the extraction is relatively simple. In order to make the scatter plot show the separation of new ice, ice rind, and seawater better, first only the sample points of new ice, ice rind, and seawater were selected to construct the scatter plot. The optimal band combination was selected by averaging the Euclidean distance and inter-class variance combined with visual interpretation.

**Table 2.** First band combination list.

| R | G | R + G | R + B | R + NIR | G + B | G + NIR |
|---|---|---|---|---|---|---|
| B | NIR | R * G | R * B | R * NIR | G * B | G * NIR |
| B + NIR | R − G | R − B | R − NIR | G − B | G − NIR | B − NIR |
| B * NIR | R/G | R/B | R/NIR | G/B | G/NIR | B/NIR |

The average Euclidean distance is

$$x = \frac{1}{n} \sum_{i=1}^{n} x_i, \tag{1}$$

where $x_{(y)}$ is the average Euclidean distance of the sea ice or seawater on the $x$-axis or $y$-axis, $n$ is the number of sea ice or seawater sample points, and $x_i$ is the reflectivity of sea ice or seawater.

$$U = \sqrt{(x_{\text{ice}} - y_{\text{ice}})^2 + (x_{\text{sea}} - y_{\text{sea}})^2}, \tag{2}$$

where U is the average Euclidean distance between the sea ice and seawater in two-dimensional space, $x$ and $y$ are the average Euclidean distances of the sea ice sample points on the $x$-axis and $y$-axis, respectively, and $x$ and $y$ are the sea ice sample points on the $x$-axis and $y$-axis, respectively.

The variance between classes is

$$\sigma = \sum_{i=1}^{n} (x_i - \bar{x})^2, \tag{3}$$

where σ is the between-class variance of sea ice or seawater, and xi is the number of sample points of sea ice or seawater. The larger the average Euclidean distance and the smaller the variance between classes, the higher the degree of separation is.

According to the principle of a larger average Euclidean distance and a smaller variance between classes, the top 12 band combinations were selected and their scatter plots were examined. As shown in Figure 6, the abscissas and ordinates of the scatter plots are the reflectance of the band after a band combination, for example, B is the reflectance of the green band, and R/B is the ratio of the reflectance of the red band to the green band. It was found that in the scatter plots of B and R/B, sea ice and seawater can basically be separated by a straight line. The mixing of seawater is more serious. Therefore, the band combination with the best reflectivity ratio between the green band and the red band was finally selected.

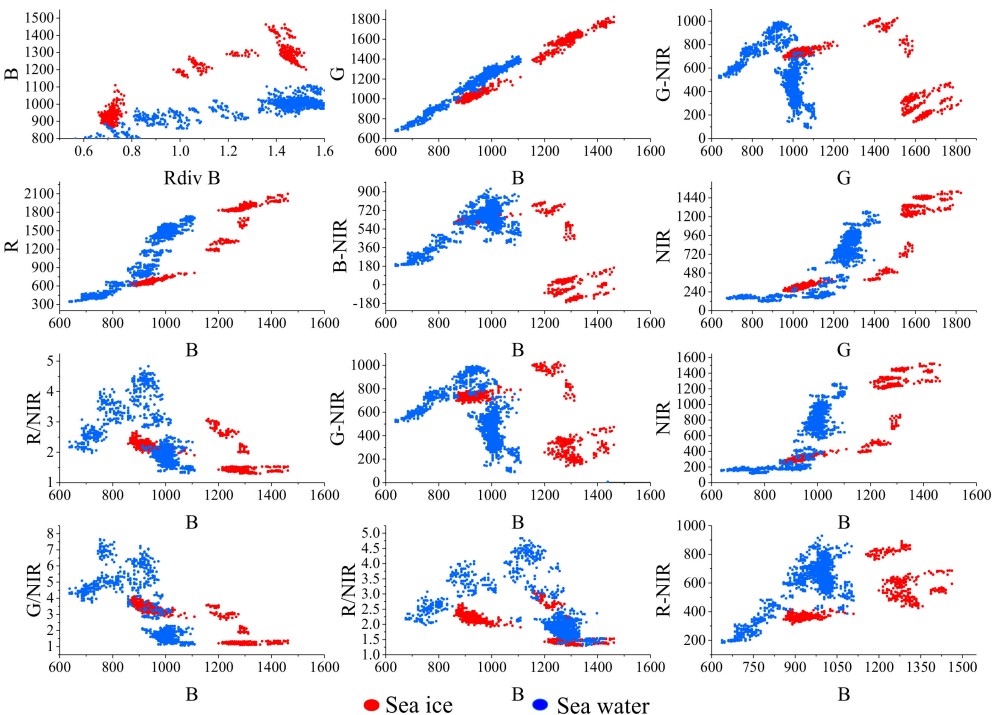

**Figure 6.** Scatter plots of band combinations for seawater and sea ice.

R/B and B were used as the *x*-axis and *y*-axis, respectively, to draw the scatter plots of the seawater and different ice types (Figure 7). The seawater reflectance samples exhibit good linearity in the scatter plots. The different types of sea ice are located above the straight line. Therefore, the linear equation of seawater was obtained through linear fitting as the dividing line.

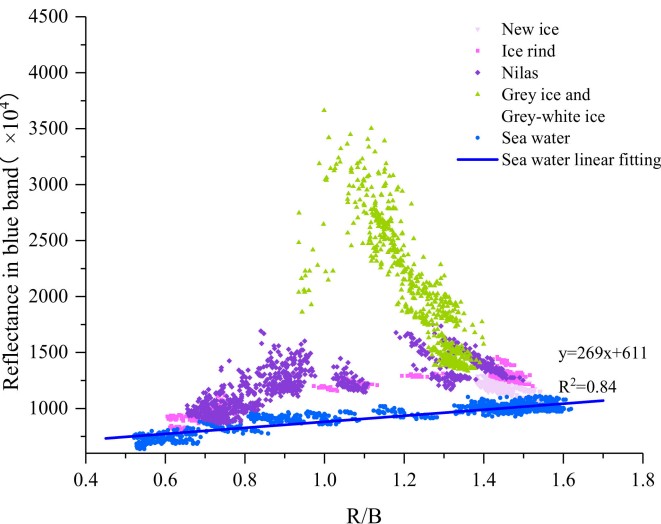

**Figure 7.** Scatter plot of the reflectivity of the different types of sea ice and seawater.

The dividing line equation is

$$y = 269x + 611,\tag{4}$$

where x is the reflectance ratio of the red band to the green band, and y is the reflectance of the green band.

In order to make the value of the sea ice a positive number to facilitate subsequent statistical analysis and the determination of the threshold, only the slope of the seawater linear fitting line was used, and finally the sea ice spectral information index was constructed:

$$y = B - 269 \times (R/B),\tag{5}$$

where y is the calculated reflectance value of each pixel in the image (sea ice has a larger value and seawater has a relatively small value); B is the reflectance value of the green band; and R/B is the ratio of the reflectance in the red band to that in the green band.

### 2.4. Sea Ice Spatial Information Extraction

As shown in Figure 8, in optical images, such as GF1, Landsat, and Sentinel-2 images, some thin ice such as new ice and ice rind exhibits spectra very similar to that of seawater containing suspended particles. Sea ice in seawater with a higher concentration of suspended particulate matter cannot be extracted using spectral information alone. Therefore, it is necessary to distinguish this part of the sea ice from the seawater based on the spatial information concerning the sea ice. Compared with the smooth spatial characteristics of the seawater surface, the surface of sea ice is generally rough, with conspicuous irregular and unstable textural characteristics. In this section, three different spatial information extraction schemes are designed to explore the applicability of the textural features and edge features of the gray-level co-occurrence matrix to the extraction of various types of ice; and through comparison of these schemes, the best method for extracting sea ice spatial information is determined.

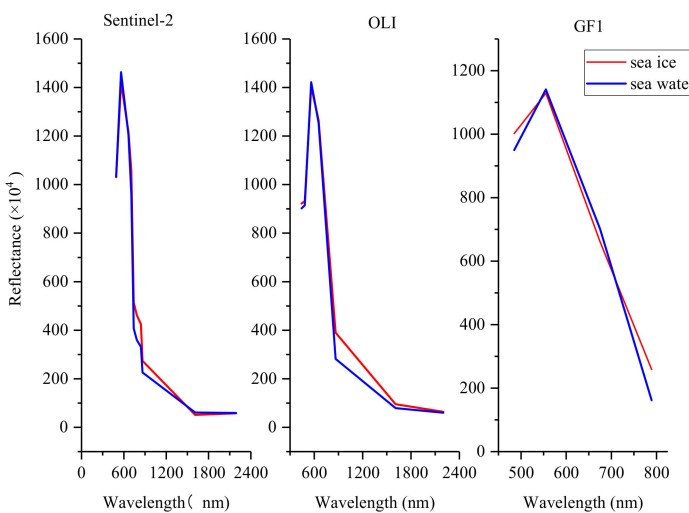

**Figure 8.** The similarity of the sea ice and seawater spectral curves for the different sensors.

The textural features represent the surface conditions of the object, such as smooth or rough, which helps to distinguish homogeneous and heterogeneous regions. Since sea ice has irregular and unstable textural characteristics, the textural characteristics of the image were added when the sea ice was extracted in order to solve the problem of sea ice and seawater bearing similar spectra. At present, the commonly used method of extracting sea ice textural features is the gray-level co-occurrence matrix method. The gray-level co-occurrence matrix is a matrix that counts the gray-level relationship between pixels within a certain interval in a local area of an image. The factors affecting the gray level co-occurrence matrix are the image quantization level, the size of the moving window, the movement direction, and the movement step length. The gray-level co-occurrence matrix provides information about the image's gray direction, interval, and change range. Based on the gray-level co-occurrence matrix, the statistical attributes that quantitatively describe the textural features are extracted. Haralick et al. (1973) defined 14 textural features [38]. The feature statistics commonly used to extract textural information from remote sensing images mainly include the mean, variance, homogeneity, contrast, dissimilarity, entropy, angular second moment, and correlation. Recently, many related studies have been conducted on the extraction of sea ice information based on the textural features of the gray level co-occurrence matrix, but there remains a lack of research on the applicability of textural features to various types of ice. Therefore, the applicability and advantages of the different textural features in extracting sea ice types were explored.

Sea ice has conspicuous edge characteristics under different SPM contents. The Sobel edge detection operator has the advantage of easy calculations and a strong anti-noise ability. The edge detection image value of the Sobel operator represents the gradient value of the pixels in the region, and the edge of the sea ice has a higher gradient value. Therefore, the edge points of the sea ice were extracted using the Sobel operator, and the edge point density map is generated with the number of sea ice edge points in a certain range. The edge point density map represents the density of the sea ice edges in a local area. The higher the sea ice density, the greater the number of sea ice edge points. The edge point density map was used to explore the edge characteristics of the sea ice.

In order to further explore the best scheme for extracting the spatial information about the sea ice and to delve deeper into the sea ice spatial information, Scheme 3 combines the edge point density map and the statistics of each textural feature through multiplication, and further explores the ability to combine the edge and textural features to extract the sea ice.

### 2.5. Object-Oriented Extraction of Sea Ice Extent

Object-oriented classification refers to the segmentation of images to form objects with adjacent homogeneous pixels, which overcomes the limitations of traditional remote sensing image classification methods that use pixels as the basic classification and processing unit to contain more semantic information [39]. The object is the processing unit, which can achieve a higher level of remote sensing image classification. The object-oriented remote sensing image classification method is not only based on the spectral features but also uses the textural features of the image to segment and classify the image. The classification results avoid speckle noise and have good integrity. Image segmentation is the most basic and critical step in the object-oriented classification method, which directly determines the accuracy of the classification results and the workload of the classification process. In order to improve the accuracy and efficiency of the segmentation, in this paper study, the edge detection segmentation algorithm and the full lambda schedule merge algorithm in ENVI were used.

The blue band is less affected by a high SPM content, and the texture and edge information about the sea ice is clearer. Therefore, the blue band of the image was selected as the reference image to segment the sea ice spectral information and the spatial information in the image. The mean attribute of the segmented object was used to extract the sea ice. Figure 9 shows that the object-oriented method can not only reduce the speckle noise in the classification results, but also limit the influence of the spatial feature window factor when using the spatial features to extract the sea ice and can improve the accuracy of the classification results.

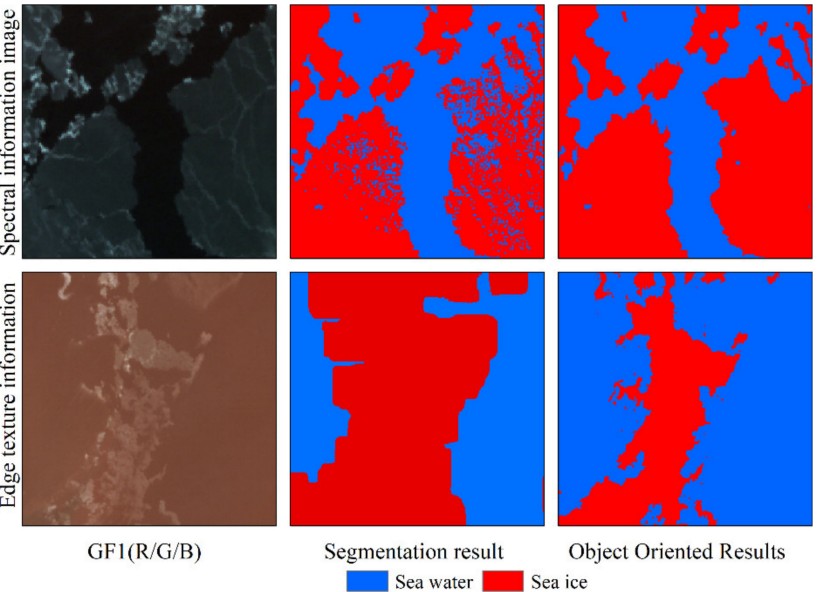

**Figure 9.** Object-oriented segmentation results.

### 2.6. Determination of Segmentation Threshold Based on OTSU

The automatic determination of the object-oriented segmentation threshold affects the final classification result and the automatic process of sea ice range extraction. In this study, the OTSU method was used to automatically determine the threshold. The principle of the OTSU method is to continuously iteratively determine an optimal threshold to maximize the variance between the target and the background. Before conducting the OTSU threshold segmentation, the terrestrial mask pixels need to be removed. This is because the OTSU determines the segmentation threshold based on histogram statistics. Land pixels will affect the structure of the histogram and cause the predicted threshold to deviate. After removing the land pixels, the double peaks in the histogram are clearer. This improves the accuracy of the threshold.

### 2.7. Accuracy Verification

In order to better evaluate the robustness and applicability of the method developed in this study, the proposed method was compared with the extraction results of the Support Vector machine (SVM) and K-Means methods, and the three methods were applied to GF1, Landsat-8, and Sentinel-2 images. In order to quantitatively evaluate the accuracy of the sea ice extraction, ArcGIS was used to randomly generate 800 test points in the sea area, and the type was marked based on a planet satellite image with a resolution of 3 m. To ensure that the test points were evenly distributed in the study area and that all types of sea ice and seawater were present, their total accuracy and kappa coefficient ($\kappa$) were calculated.

## 3. Results

### 3.1. Analysis of Sea Ice Spectral Information Index

Based on a GF1 image acquired on 12 January 2018, 800 sea ice and seawater sample points were selected, and the sea ice spectral information index was used to plot the distribution ranges of the different types of sea ice and seawater reflectance values (Figure 10). The results revealed that after the sea ice spectral information index was constructed, the reflectance values of the different types of sea ice were larger, while the reflectance values of the seawater were concentrated within a small interval, indicating that the sea ice spectral information index can effectively extract the sea ice spectral information.

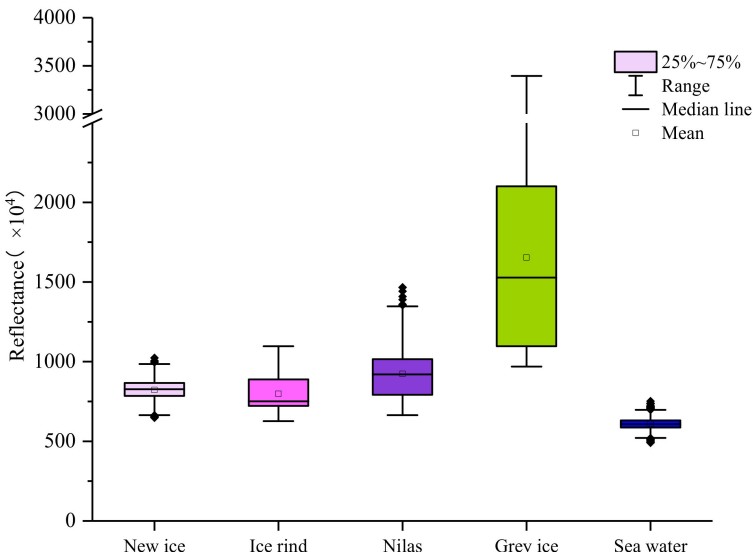

**Figure 10.** The range of sea ice spectral information index of sea ice and sea water.

The sea ice can be initially extracted by selecting a suitable threshold. Figure 11 shows the sea ice extraction results. It can be seen that the sea ice spectral information index can effectively extract the sea ice in seawater with different suspended sediment concentrations. The new ice and ice rind in the high suspended sediment area can also be extracted more accurately. However, there are still some problems in the classification results. First, there is the salt and pepper phenomenon, which is a common problem in pixel-based classification methods. This will be solved by object segmentation and extraction. Second, there is still a small amount of confusion between seawater and sea ice in area c, which is mainly concentrated in the areas where the concentration of suspended particles changes drastically. This is because the seawater in these areas display spectral curves that are extremely similar to those of some of the types of sea ice such as new ice and ice rind, and this phenomenon is present in the GF1, Landsat, and Sentinel-2 images (Figure 8). Therefore, it is not possible to completely distinguish between sea ice and seawater using only the spectral characteristics of the image, thus necessitating the addition of the spatial characteristics of the image.

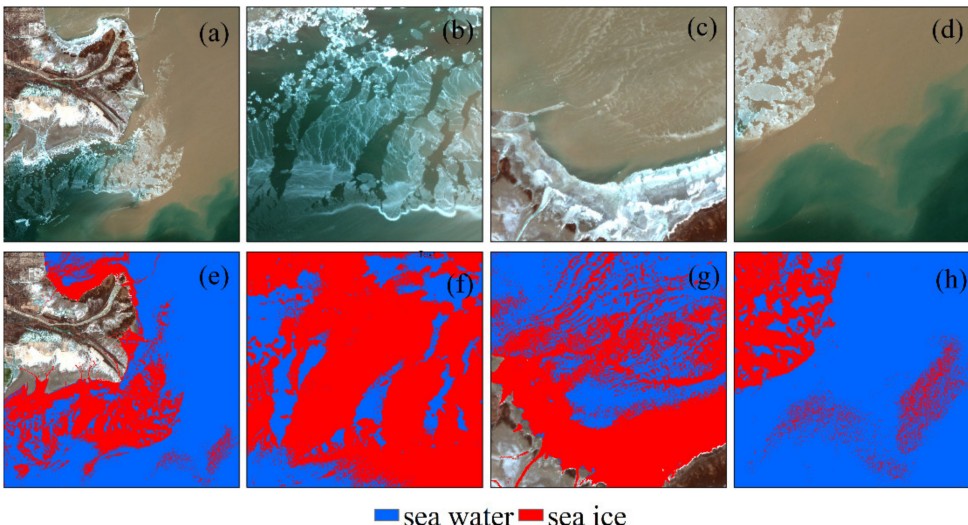

**Figure 11.** Sea ice extraction results using spectral information. (**a**) GF1 image (R/G/B); (**b–d**) The three sub-areas of the study area; (**e–h**) The extraction results using spectral information, respectively.

### 3.2. Optimization of Spatial Feature Extraction Scheme

Scheme 1: The textural feature parameters based on the gray-level co-occurrence matrix mainly include the quantization level, the size of the moving window, and the movement direction and step length. Since the directional characteristics of sea ice are not evident, default values (0,1) were used for the movement directions of the x-axis and y-axis. Moreover, the movement step length was set to a default value of 1. The following section only discusses the quantization level of the gray-level co-occurrence matrix and the moving window size in detail.

Without compressing the gray level of the original image, the size of the gray level co-occurrence matrix is the square of the gray level of the original image, which will greatly increase the calculation load of the gray level co-occurrence matrix. Therefore, in practical applications, in order to improve the efficiency of the calculation of the textural features, the gray level of the original image is usually compressed, and quantization levels of 64, 32, and 16 are generally used.

Figure 12 shows the characteristics of the sea ice and seawater in the GF1 images under different quantitative levels. It can be observed that the images with 64 quantization levels maintain the textural characteristics of the original images better; the images with 32 quantization levels display a reduced ability to maintain details; and the images with 16 quantization levels have lost a significant amount of textural information. Therefore, the higher the quantization level, the better the textural details of the original image are preserved. However, images with high quantization levels are not suitable for the extraction of sea ice textural information. Figure 13 shows the four textural feature indexes of the homogeneity, dissimilarity, entropy, and second moment under different quantization levels. Due to the drastic changes in the concentration of the suspended sediment in the Yellow River Delta, the images with 64 quantization levels exhibit a large amount of speckle noise in the seawater area. In the 32 quantization level images, this speckle noise is greatly suppressed. In addition, since the calculation load increases with increasing quantization level, the calculation efficiency is lower. Therefore, the quantization level of the gray-level co-occurrence matrix was finally set to 32.

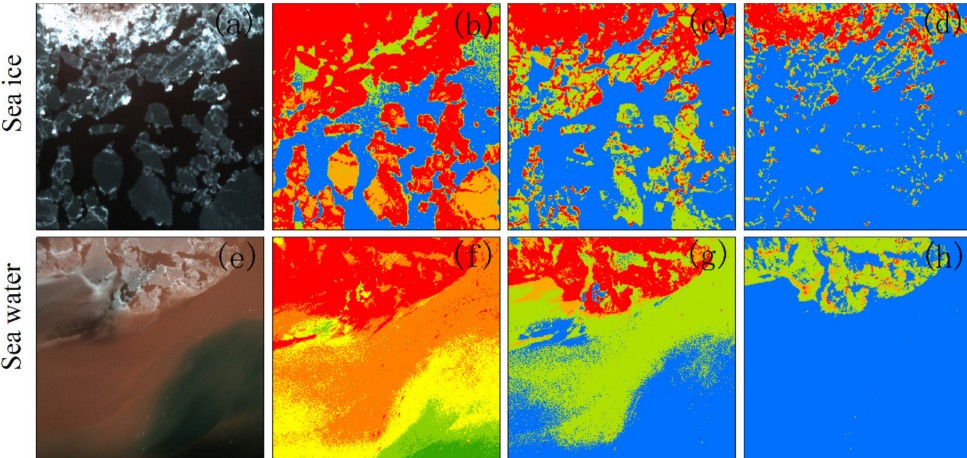

**Figure 12.** Image features at different quantization levels. (**a**) Sea ice areas in GF1 images; (**b**–**d**) sea ice images at 64, 32, and 16 quantization levels, respectively; (e) Sea water areas in GF1 images; (**f**–**h**) sea water images at 64, 32, and 16 quantization levels, respectively.

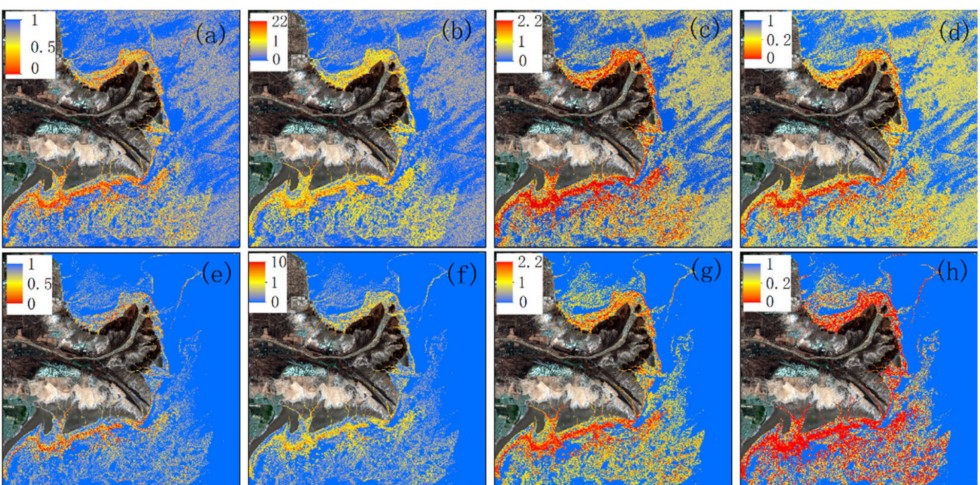

**Figure 13.** Images with different textural feature parameters under different quantization levels. (**a**–**d**) Texture image of homogeneity, dissimilarity, entropy, second moment at 64 quantization levels; (**e**–**h**) Texture image of homogeneity, dissimilarity, entropy, second moment at 32 quantization levels.

The moving window is an important factor that affects the textural feature extraction of the gray-level co-occurrence matrix. Figure 14 shows the distribution range of the textural feature values of various types of sea ice and seawater for different window sizes. It can be seen that the size of the window has little effect on the textural characteristics of the sea ice and seawater, but as the texture window increases, the calculation load increases greatly, thus the window size selected in this study was 3. Based on the statistical results of the textural feature index values of the various types of sea ice and seawater, grey ice and grey-white ice have a higher degree of discrimination from seawater in terms of each textural feature value. The types of thin ice such as new ice, ice rind, and nilas cannot be completely distinguished from the textural characteristics of seawater. This is because the surfaces of the ice rind and nilas are relatively smooth, which is similar to the textural characteristic value of seawater. The surfaces of grey ice and grey-white ice are rough, and the textural characteristic value of seawater is quite different.

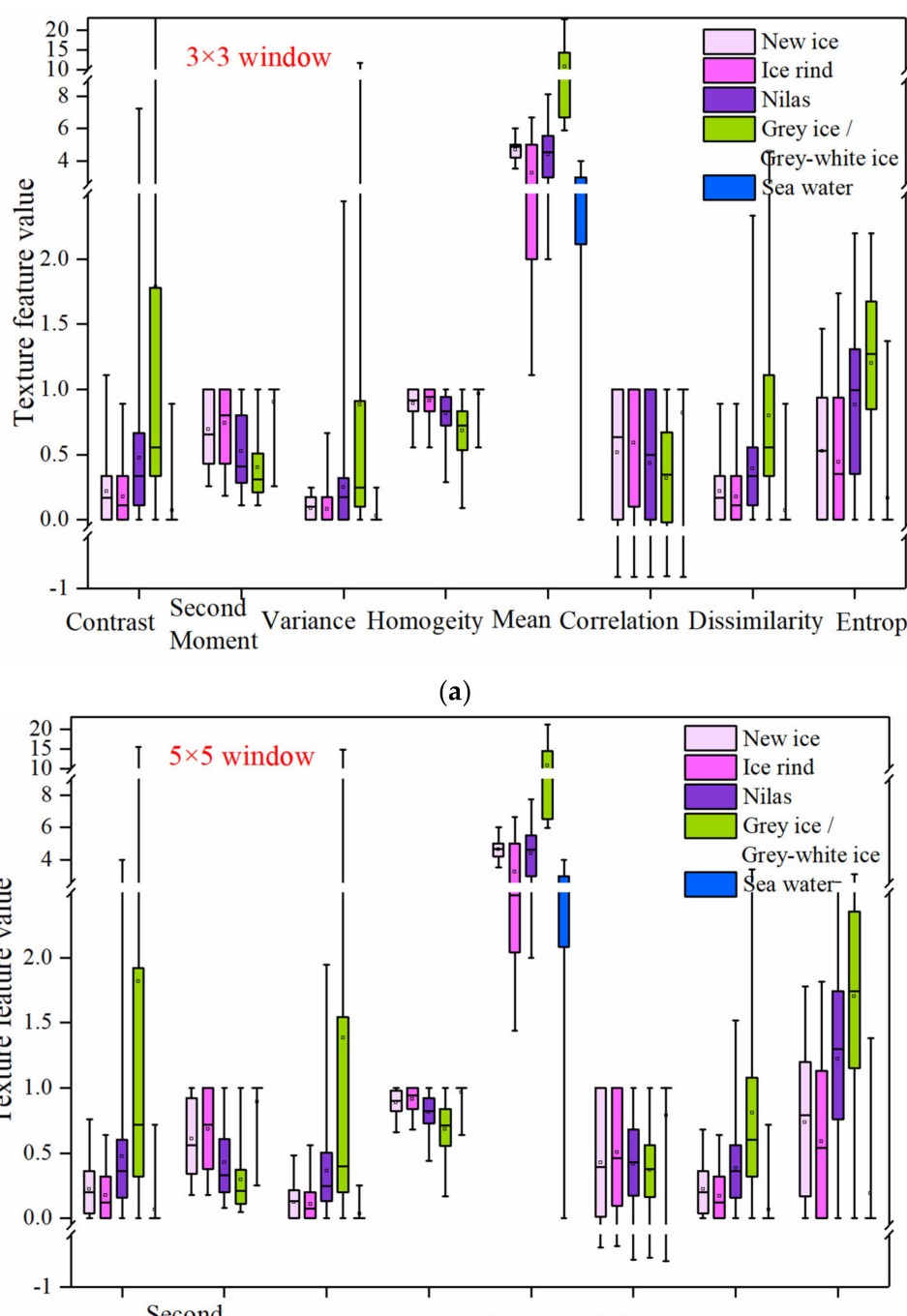

**Figure 14.** *Cont.*

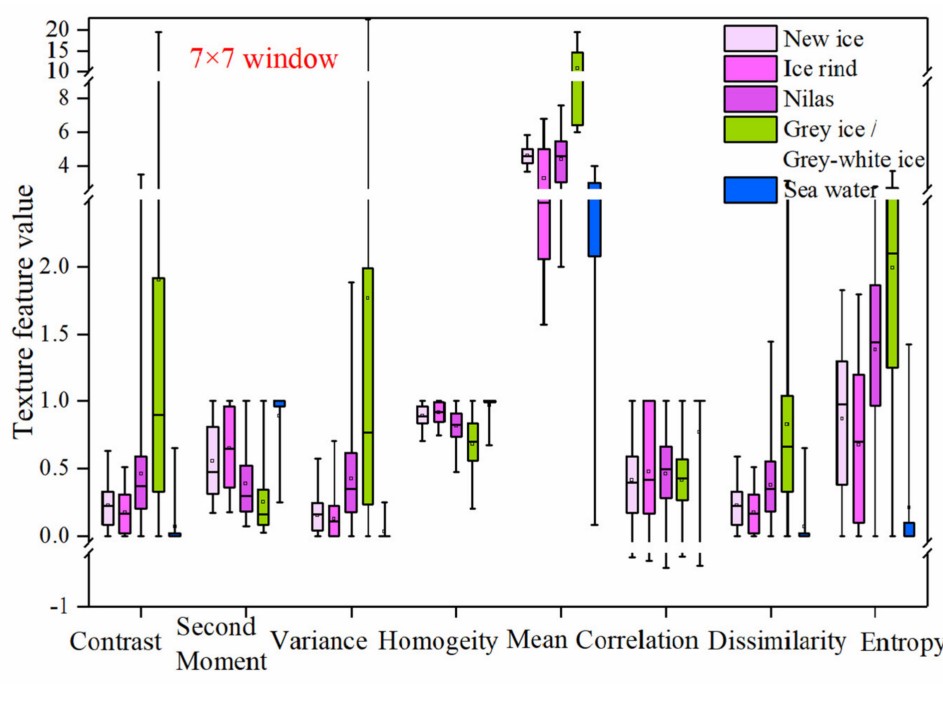

(**c**)

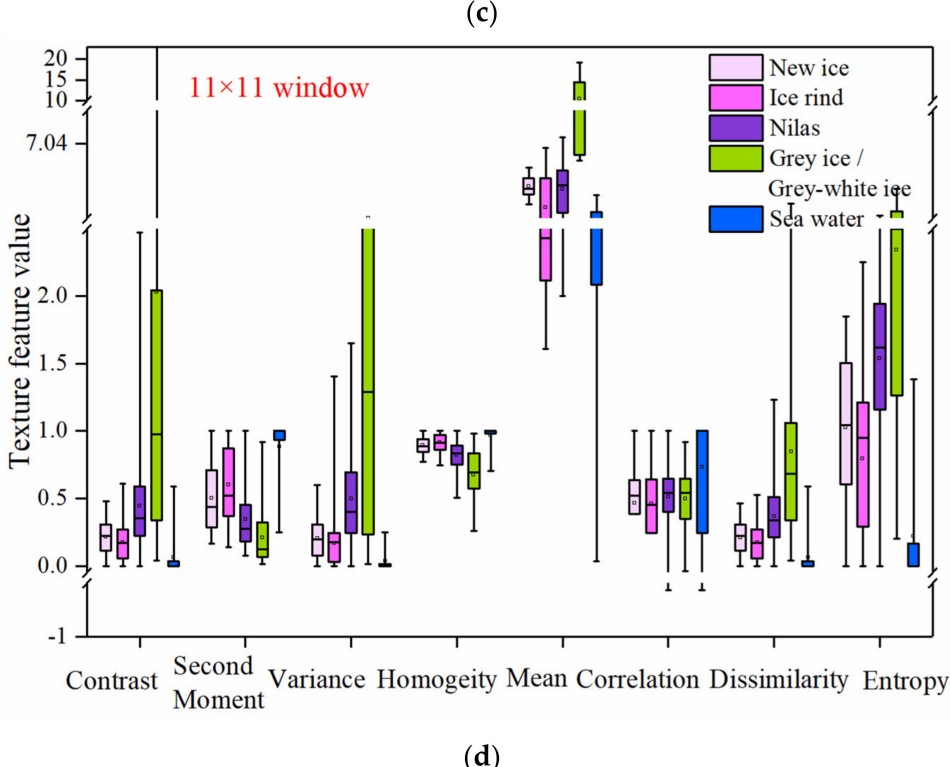

(**d**)

**Figure 14.** Plots of the ice water textural characteristic indicators for different window sizes. (**a**) Ice and water texture value distribution in 3 window sizes; (**b**) Ice and water texture value distribution in 5 window sizes; (**c**) Ice and water texture value distribution in 7 window sizes; (**d**) Ice and water texture value distribution in 11 window sizes.

Scheme 2 uses the Sobel operator to generate an edge point density map to highlight the edge features of the sea ice. The distribution ranges of the edge density values of the different types of sea ice and the seawater for different window sizes were calculated (Figure 15), and the optimal calculation window size for the sea ice edge points was compared and analyzed.

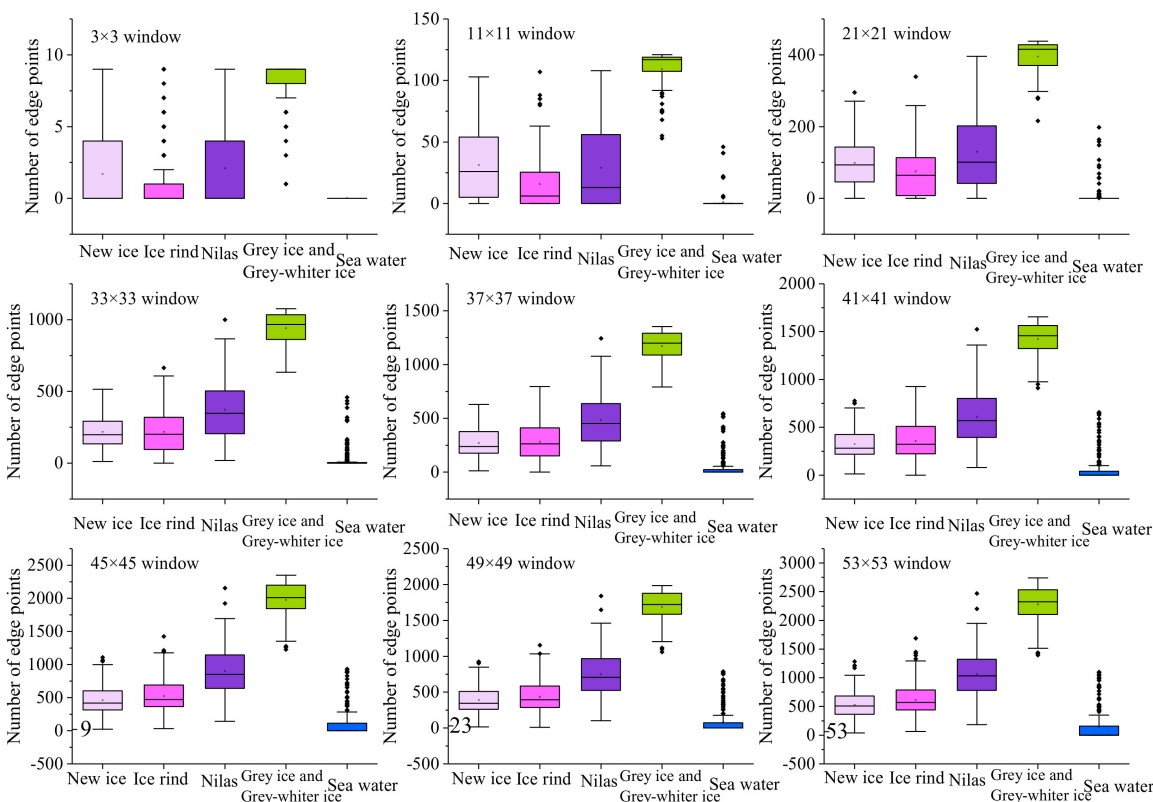

**Figure 15.** The effect of the window size on sea ice extraction using an edge point density map.

When the window was small, the edge density value of the seawater basically approached 0, and the edge density values of the grey ice, grey-white ice, and seawater were significantly different. The edge density values of the new ice, ice rind, nilas, and seawater partially overlapped. The overlapping area mainly contained the inner smooth sea ice. As the window size increased, the number of edge points that were detected inside the thin ice region such as new ice, ice rind, and nilas increased, and the edge point density value gradually increased. When the window size reached 45, the edge point density values of the various types of sea ice were significantly different from those of the seawater. As the window continued to grow, it greatly increased the amount of calculation load, thus 45 was selected as the best window size.

In Scheme 3, the texture feature window size was set to 3 × 3, and the quantization level was set to 32. After the edge point density map was combined with the various textural feature indicators, the distribution ranges of the various types of sea ice and the seawater were determined (Figure 16). It can be seen from Figure 16 that the combination of textural feature indicators such as the variance, homogeneity, and contrast with the edge point density map failed to produce a better extraction effect. After the mean textural feature was combined with the edge point density map, the range of the seawater decreased further and became more concentrated, and the distinction between the various types of sea ice and the seawater increased further. Therefore, the edge point density map combined with the mean textural feature index was selected as the final solution for extracting the sea ice spatial information.

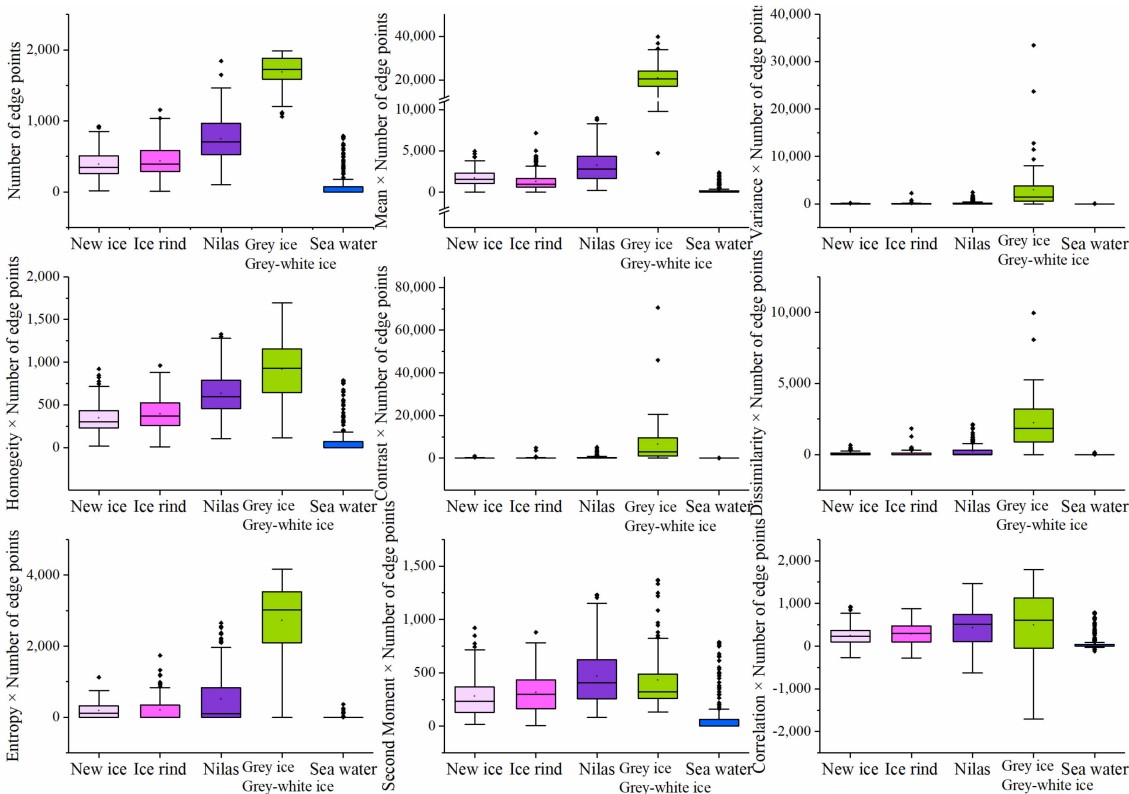

**Figure 16.** Box plots for the combinations of the edge density map and textural feature.

Figure 17 shows the comparison between the edge texture information extraction results and the spectral information extraction results. The edge texture image can extract the extent of the sea ice as a whole and can extract the types of ice such as new ice, nilas, grey ice, and grey-white ice. The most important factor is that the texture images at the edges can compensate for the similarity between the spectra of the sea ice and the seawater. As shown in Figure 17j,o, the extraction accuracy of the spectral information is lower in areas where the concentration of the suspended particulate matter changes drastically. The edge texture images solve this problem. Although the seawater in the crevices between portions of ice can also be identified as sea ice, it can be combined with the spectral information to achieve a more accurate sea ice extraction.

### 3.3. Accuracy Verification

Figure 18 shows the sea ice extraction results obtained using the different methods for a GF1 image acquired on 12 January 2018. Four scenes including new ice, ice rind, nilas, grey ice, and grey-white ice were selected to illustrate the results of the sea ice extraction. In addition, the results were compared with the sea ice extraction results obtained using the K-Means and SVM methods. Taking into account the complexity of the changes between the various types of sea ice in the seawater with different suspended particulate matter concentrations in the Yellow River Delta, in order to improve the accuracy of the K-Means and SVM methods as much as possible, the K-Means method categories were set to 2–10 categories, and then, the classification post-processing was performed. The image was finally divided into two categories, namely, sea ice and seawater. When the SVM method was employed to select the sample points, the sample points were selected according to the types of ice in the Yellow River delta, turbid seawater, and clear seawater in order to improve the accuracy of the sea ice extraction.

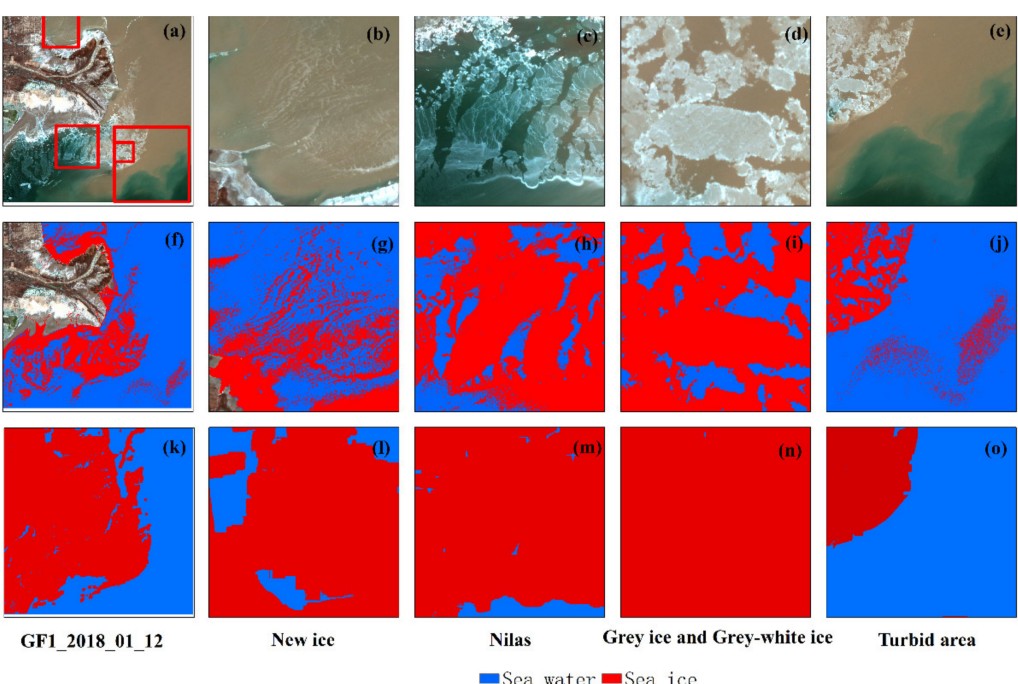

**Figure 17.** Comparison of edge texture results and spectral results. (**a**–**e**) GF1 true color images; (**f**–**j**) results of sea ice extraction from spectral information; and (**k**–**o**) results of the sea ice extraction from edge texture information.

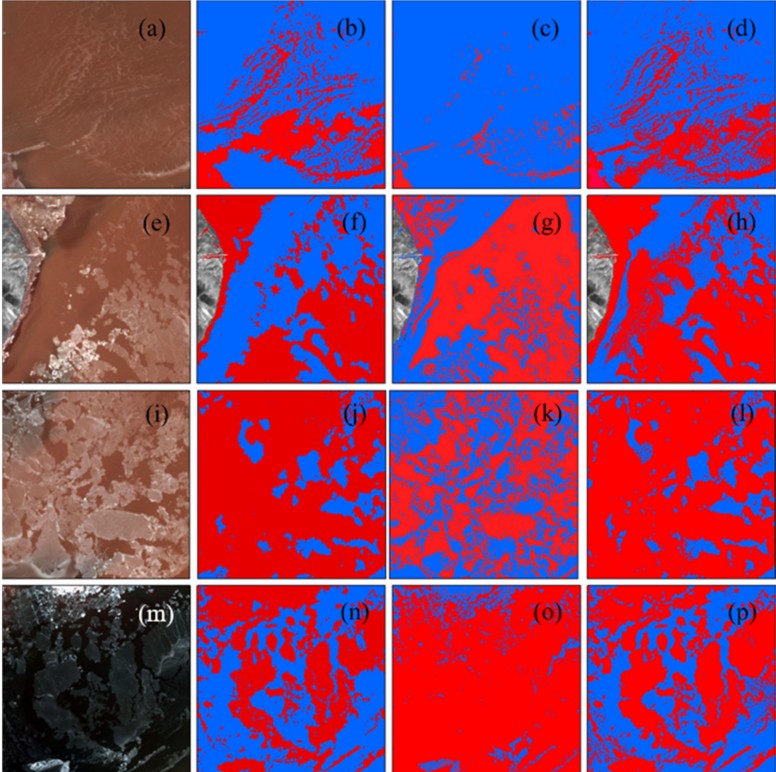

**Figure 18.** Comparison of the sea ice extraction results obtained using different methods. (**a**,**e**,**i**,**m**) True color images of the GF1 image acquired on 12 January 2018; (**b**,**f**,**j**,**n**) Classification results for the method proposed in this paper; (**c**,**g**,**k**,**o**) K-Means classification results; (**d**,**h**,**l**,**p**) SVM classification results.

It can be observed from Figure 18c,h that the K-Means method cannot completely extract the sea ice when extracting thin ice such as new ice and ice rind in seawater with a high suspended particulate matter concentration. As Figure 18g,o shows, most of the seawater was classified as sea ice in the areas with high suspended particulate matter concentrations near the shore and in the clear water areas. This demonstrates that the K-Means method is greatly affected by suspended sediment. The results of the SVM method of extracting sea ice were generally better than those of the K-Means method, but most of the seawater remained classified as sea ice in the areas with high suspended particulate matter concentrations. In addition, there is a significant salt and pepper phenomenon present in the extraction results. The method proposed in this paper can accurately extract the various types of sea ice in both turbid seas and clean seas. It also greatly reduces the salt and pepper phenomenon and improves the integrity of the sea ice extraction.

In order to quantitatively evaluate the accuracy of the sea ice extraction, the overall accuracies and kappa coefficients of the classification results for the GF1, Landsat 8, and Sentinel-2 images were compared and analyzed and additionally compared with those of the K-Means and SVM methods. In addition, the method was applied to the Yellow River Delta and Liaodong Bay. The results are presented in Table 3. The overall accuracy of the method proposed in this paper is basically >95%, the kappa coefficient is > 80%, and the accuracy is 5% higher than those of the SVM and K-Means methods. On 21 January 2017, there were mixed pixels of clouds and water in some areas, which affected the accuracy of the final sea ice extraction. In Liaodong Bay, the accuracy of the SVM was close to that of the method proposed in this paper. This is because the sea ice in Liaodong Bay is predominantly thick ice such as grey ice and grey-white ice, and is less affected by suspended sediment. Therefore, both the proposed method and the SVM method achieved better accuracies.

**Table 3.** Accuracy evaluation table.

| Area | Date | Image | Method | OA | k |
|---|---|---|---|---|---|
| Yellow River Delta | 12 January 2018 | GF1 | This method | 0.98 | 0.96 |
| | | GF1 | SVM | 0.93 | 0.86 |
| | | GF1 | K-Means | 0.78 | 0.55 |
| | 21 January 2017 | GF1 | This method | 0.93 | 0.81 |
| | | GF1 | SVM | 0.84 | 0.59 |
| | | GF1 | K-Means | 0.77 | 0.45 |
| | 12 January 2018 | Sentinel-2 | This method | 0.99 | 0.98 |
| | | Sentinel-2 | SVM | 0.9 | 0.95 |
| | | Sentinel-2 | K-Means | 0.81 | 0.60 |
| | 23 January 2019 | Landsat-8 | This method | 0.94 | 0.88 |
| | | Landsat-8 | SVM | 0.89 | 0.77 |
| | | Landsat-8 | K-Means | 0.76 | 0.46 |
| Liaodong Bay | 17 February 2019 | Landsat-8 | This method | 0.99 | 0.98 |
| | | Landsat-8 | SVM | 0.96 | 0.95 |
| | | Landsat-8 | K-Means | 0.91 | 0.82 |

The final results show that the accuracy of the K-Means method was the lowest among the three methods. This is due to the similarity between the spectra of the highly turbid seawater and thin ice sheets in the Yellow River Delta and the complexity of the various types of sea ice in the different turbid seawater regions. This led to the relatively low classification accuracy of the K-Means method. The SVM method exhibited a better classification accuracy than the K-Means method overall, but it only used the spectral information, thus the classification accuracy of the ice types, such as in the high suspended sediment areas and for ice rind, was lower. In addition, the SVM method is reliant on prior knowledge. It is a time-consuming process to manually select sample points, and the quality of the sample points directly affects the accuracy of the final classification. The method proposed in this paper attained good accuracy in both the turbid water and clear

water areas, and achieved automation of the sea ice extraction. All processing methods were carried out in ENVI. The ENVI functions are called using IDL and can be easily automated.

## 4. Discussion

In recent years, extreme weather such as high temperatures, droughts and floods have occurred frequently, and climate anomalies have become the norm, which has led to people's cognitive thinking on global climate change and human living environment [40,41]. As an indicator of global climate change, sea ice change is related to global warming, rises in sea levels and other issues [42,43]. The development of ice conditions in the Yellow River Delta waters in China is unstable, and the formation of sea ice is rapid, which responds more closely to local regional climates. Accurate monitoring of sea ice extent is therefore crucial. Suspended sediment in the mouth of the Yellow River significantly affects the accuracy of sea ice extent extraction. This paper proposes an automatic extraction method of sea ice that combines texture, edge and spectral information, which improves the accuracy of sea ice extraction under highly dynamic suspended sediment changes. Compared with SVM and K-Means, the accuracy is improved by more than 5%. This method provides a basis for accurate sea ice identification using GF1 images, and also offers a method for other optical remote sensing data. High-resolution satellite data based on multiple sources can compensate for the lack of data time resolution and further improve its sea ice monitoring capabilities. Therefore, sea ice monitoring based on multi-source remote sensing data will be the key direction of future development. Moreover, this method provides an approach for other optical remote sensing data, which is of great significance for making full use of multi-source remote sensing data to study the law of sea ice change. Accurate identification of sea ice extent is of great significance to sea ice monitoring, sea ice prediction, disaster prevention and mitigation, and climate research in the Yellow River Delta region. Although this paper discusses the characteristics of various sea ices in detail and enables higher-precision sea ice extraction, it does not distinguish between various sea ice types. Accurate identification of sea ice types is of great significance to the study of sea ice production, ablation and migration. Most of the sea ice in the Yellow River Delta is less than 30 cm thick, and it remains difficult to classify them with greater precision. In addition, the spectrum, texture, and edge information of coastal ice and floes such as grey and white ice are relatively close, and it is difficult to distinguish between coastal ice and floating ice. Therefore, in the future, we will study the distinction of various sea ice types and realize the identification methods of different types of sea ice.

## 5. Conclusions

The automatic and accurate extraction of sea ice is essential for studying the laws of sea ice generation and migration, improving sea ice disaster prevention and mitigation, and monitoring climate change. Accurate real-time observations of sea ice bear an important application value and is of theoretical significance.

In order to solve the problem of the low sea ice extraction accuracy caused by the influence of the suspended sediment in the Yellow River Delta, in this study, an automatic sea ice extraction method combining sea ice spectral, texture and edge information is proposed, where the sea ice extraction accuracy can reach over 93%, which is more than 5% higher than SVM and K-means. Compared with previous studies, the sea ice spectral information index suitable for different suspended sediment concentrations is constructed by a two-dimensional scatter diagram of characteristic bands, which improves the applicability of sea ice spectral information index. In changing from discussing the texture characteristics of sea ice as a whole in the past, this study discusses the texture characteristics and edge characteristics of various sea ice types in the Yellow River Delta in detail, laying a foundation for the classification of sea ice types. In addition, the automatic determination of threshold based on OTSU can realize the automatic extraction of sea ice. The method in this paper uses only four bands of visible light and near-infrared to extract sea ice, thus providing a method to be extended to other high-resolution optical remote

sensing data and is of great significance to maximally utilize multi-source remote sensing data for real-time monitoring of sea ice.

In future research, we may expand the research area to the Bohai Sea in China, and realize real-time observation of sea ice through Landsat, Sentinel-2, GF1 and other optical remote sensing data. In terms of data sources, in order to improve the frequency of sea ice monitoring, SAR data may also be applied. We hope to conduct high-precision and high-frequency sea ice monitoring, so as to make a certain contribution to preventing disasters and studying climate change around the Bohai Sea.

**Author Contributions:** Conceptualization, H.Q. and Z.G.; methodology, H.Q.; software, K.M.; validation, H.Q., J.H. and Y.K.; formal analysis, D.Z.; investigation, Z.G. and Y.K.; resources, H.Q.; data curation, H.Q.; writing—original draft preparation, H.Q.; writing—review and editing, H.Q.; visualization, H.Q.; supervision, H.Q.; project administration, H.Q.; funding acquisition, Z.G. All authors have read and agreed to the published version of the manuscript.

**Funding:** This research was funded by National Natural Science Foundation of China grant numbers 41971381 and 42071396. This research was funded by National Key R&D Program of China grant number 2017YFC0505903.

**Institutional Review Board Statement:** Not applicable.

**Informed Consent Statement:** Not applicable.

**Data Availability Statement:** Not applicable.

**Conflicts of Interest:** The authors declare no conflict of interest.

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
