# Peer review of "Automatic and Accurate Extraction of Sea Ice in the Turbid Waters of the Yellow River Estuary Based on Image Spectral and Spatial Information"

_remotesensing, doi:10.3390/rs14040927_

Round 1

Reviewer 1 Report

General remarks:

The major issue of this paper is the lack of a discussion section. I strongly suggest supplementing the following parts of the discussion with proper citations:

  • Explanation of the main discoveries and speculation about them (supported by reference to literature)
  • Implications of the research (in relation to the Results: what they mean for the broader field of research)
  • Limitations of your study
  • Recommendations for future research

Specific comments:

line 52: can you convert the currency to US dollars? It would be much more clear for the potential readers

line 56: can you explain or provide more details of the "new era" that is mentioned in the text?

line 63: what do you mean by 'precise extraction of the sea ice'? Extraction is typically associated with mining, which is probably not the purpose of your article. Please be more consistent.

line 72: what kind of data source? Please clarify.

line 74: SAR data band? Please clarify.

line 76: unnecessary repetition (interpretation)

lines 88-90. This sentence should be moved to the recommendations for the future research part of the discussion.

line 98. Suggested reference for GLCM:

  • 10.1109/tsmc.1973.4309314

lines 118-119. Proper references for SVM and CART:

  • 10.1007/bf00994018
  • Breiman, L., et al. (1984). Classification and Regression Trees. Belmont, Wadsworth.

line 131: explain why the process is more complicated

line 164: reference needed

line 264: add (a) to the first sentence, check the caption

line 337: from this sentence, it seems that the roughness measures from DEM, like Vertical Ruggedness Measure, might be useful in your applications. Consider discussion about this feature and potential application of DEM in the discussion section. In addition, consider the following references that utilized DEM in remote sensing glacial applications:

  • 10.1109/tgrs.2021.3091771
  • 10.1016/j.geomorph.2020.107212

lines 372-383: Provide any reference to object-based classification that you used.

Reviewer 2 Report

The article is interesting, and the subject is worthy of research. However, the execution of the article and the research as well as presentation itself requires some improvements to proceed with its publication in the journal in my opinion. For this I advise a major revision to the authors in the following points:

All aspects related to the format and presentation of the article must be carefully reviewed. There are some oversights and mismatches with the format of the journal in its material execution.

Abstract

About 70% of the abstract focus on the methodology of automatic and accurate extraction of sea ice. Unfortunately there are no information presented of achieved results for this implementation of this method. I can see only emphasized information that the method has a good reliability and robustness. Please rewrite the abstract so that it will contain not only major background, techniques, innovations but also results and errors of implemented methodology.

Introduction

Unfortunately the Introduction part is one of the weakest part of the manuscript. First of all in a number of cases, you have facts, information, ideas or methods that were not your own, which have no in-text citation. It is very important to stick a reference at the end of the paragraph. I noticed, in many lines or paragraphs there are no references cited. For example lines 36-37 “The bottom of the Yellow River Delta has a small flat slope, shallow water low salinity, as a result of which seawater is easily frozen.” Are those results of your study or a finding of some other authors? Similarly one can find numerous situations when information presented lucks references.

Furthermore the literature review is very narrow and lucks examples of extraction of sea ice from different regions of the world. Citing only 20 literature positions in such well described subject suggests that the literature review is still to be done.

Moreover the introduction is quite extensive I advise to underline the significance and innovation of the work discussed in the paper. As number of methods are presented and compared please summarize and emphasize at the end of introduction the biggest advantage of new methodology.

 Methodology and results

On the contrast to introduction the methodology and result sections are well described and presented. I would recommend though to present a bit more information about the accuracy verification.

Discussion

The manuscript lacks a discussion chapter. It is an obligatory element in a scientific paper to present a comparison of achieved results to similar studies. This should be presented both of local and if possible a more global perspective. Lack of the discussion part refers to an earlier remark presented in introduction. The whole literature review have to be performed more widely and reedited in the manuscript.

Round 2

Reviewer 1 Report

My comments were addressed properly. I would like to see this paper published.

Author Response

Thank you very much for your review comments, they were all very useful to me.

Reviewer 2 Report

According to uploaded anwsers and new manuscript version it appears that authors performed corections to all major comments.

However I find the conclusion chapter a bit unimposing. Once again, the authors should set their approach and results against different already published papers. It would be benefitial to the manuscript if discussion woudl be performed in much bigger perspective.

Discussion chapter should be also moved before conclusion section.
